# Who Is to Blame? The Appearance of Virtual Agents and the Attribution of Perceived Responsibility

**DOI:** 10.3390/s21082646

**Published:** 2021-04-09

**Authors:** Tetsuya Matsui, Atsushi Koike

**Affiliations:** 1Faculty of Robotics and Design, Osaka Institute of Technology, Osaka 530-0013, Japan; 2Department of Science and Technology, Seikei University, Tokyo 180-8633, Japan; koike@st.seikei.ac.jp

**Keywords:** human-agent interaction, virtual agent, attribution of responsibility, human-machine collaboration, trustworthiness

## Abstract

Virtual agents have been widely used in human-agent collaboration work. One important problem with human-agent collaboration is the attribution of responsibility as perceived by users. We focused on the relationship between the appearance of a virtual agent and the attribution of perceived responsibility. We conducted an experiment with five agents: an agent without an appearance, a human-like agent, a robot-like agent, a dog-like agent, and an angel-like agent. We measured the perceived agency and experience for each agent, and we conducted an experiment involving a sound-guessing game. In the game, participants listened to a sound and guessed what the sound was with an agent. At the end of the game, the game finished with failure, and the participants did not know who made the mistake, the participant or the agent. After the game, we asked the participants how they perceived the agents’ trustworthiness and to whom they attributed responsibility. As a result, participants attributed less responsibility to themselves when interacting with a robot-like agent than interacting with an angel-like robot. Furthermore, participants perceived the least trustworthiness toward the robot-like agent among all conditions. In addition, the agents’ perceived experience had a correlation with the attribution of perceived responsibility. Furthermore, the agents that made the participants feel their attribution of responsibility to be less were not trusted. These results suggest the relationship between agents’ appearance and perceived attribution of responsibility and new methods for designs in the creation of virtual agents for collaboration work.

## 1. Introduction

In recent years, virtual agents have been widely used as an interface for intelligent systems. In many fields, these agents work together with human users. For example, they were used as virtual assistants in schools [1], hospitals [2], and factories [3]. Furthermore, some virtual agents were used to support user activities while driving [4] and on trips [5]. In such human-agent collaborations, the agents’ and users’ attribution of responsibility is an important problem. The attribution of responsibility is one important factor that affects an agent’s perceived trustworthiness and users’ motivation to engage in collaborative work [6,7]. The trustworthiness of agents has been widely researched in the fields of HAI (human-agent interaction) and HRI (human-robot interaction). Kang and Gratch showed that self-disclosure between agents and users was effective at increasing the perceived trustworthiness of virtual agents [8]. Zhao et al. showed that long-term interaction with verbal and non-verbal cues could increase agents’ perceived trustworthiness [9]. Gratch et al. showed that virtual agents that respond infrequently to humans were more trusted by users than agents that respond frequently [10]. These prior pieces of work focused on interaction design. Appearance design is also an important factor affecting agents’ trustworthiness. Geven et al. showed that an agent that looked more intelligent was more trusted by users [11]. Hertzum et al. showed that virtual agents that looked natural and recognizable were trusted by users [12]. Such research showed that how virtual agents interact and appear affected their perceived trustworthiness.

In this paper, we focused on human-agent collaboration work. Schaefer et al. showed that humans misunderstanding agents’ actions was an important factor in decreasing trustworthiness in human-agent collaboration [13]. Chen et al. showed that a robot that guessed a user’s level of trust and changed its actions on the basis of its guess could increase the performance in collaborative work [14]. Maurtua et al. showed that the perception of safety was an important factor in users’ trust and motivation to engage in human-robot collaboration in factories [3]. These prior pieces of research hardly focused on how to maintain trustworthiness and motivation when collaborative work failed. Madhavan et al. showed that users lose their trust in computers more strongly than humans when they make a mistake [15]. Rossi et al. showed that error timing and error magnitude (seriousness) affected the trustworthiness of a robot [16]. Decreasing failure is important; however, avoiding decreases in trustworthiness perceived and in motivation when agents fail is also important within long-term collaboration and interaction. de Visser et al. showed that virtual agent interfaces could reduce the disappointment users feel when agents made a mistake [17]. However, what kinds of virtual agent appearances could reduce disappointment more has hardly been researched.

Many prior works showed that the appearance of the virtual agent affected the users’ perception and action [18,19,20]. From these prior works, we thought that the appearance of the virtual agent also affected the attribution of perceived responsibility. In prior works about human-agent collaboration that we cited above, this problem was hardly discussed.

These virtual agents were often used in online shops to support customers [21]. In online shops, many non-verbal ques were investigated as important factors [22,23,24]. The appearances of virtual agents may be one of these important non-verbal factor.

Furthermore, few studies have focused on the relationship between agents’ appearance and agents’ attribution of responsibility in collaboration work. As we will discuss later, the attribution of responsibility is a key factor of human-agent collaboration work.

In this paper, we focused on this point of view, and we aimed to suggest designs for the appearance of virtual agents that can avoid trustworthiness and motivation decreasing when agents fail. For this aim, we focused on the attribution of responsibility as perceived by a user when the user cannot know whether the user or the agent failed in collaborative work. For this purpose, we focused on two factors suggested by prior works that perceived by the agents’ appearance, “agency”, and “experience”.

### 1.1. Attribution of Responsibility

Attribution is a human mental process in which humans estimate the cause of events [25]. When trouble occurs, we attribute the cause to someone. This is called “defensive attribution” [26]. Jones and Nisbett suggested that humans tend to attribute inner factors to other people’s actions; however, they attribute external factors to their own actions as stated in [27], in which it was mentioned that the impression of a virtual agent affects the attribution of responsibility.

The word “responsibility” has many connotations. In this paper, we focused on “local responsibility” [28]. Local responsibility is responsibility that arises at particular times and in particular situations [28]. We focused on responsibility for when there are only fails in human-agent collaboration and users cannot decide who was the cause of failure. Reeves and Nass suggested that virtual agents were perceived as real humans during the interaction [29]. Thus, the notion of the attribution of responsibility can be applied to HAI and human-agent collaboration work.

Many prior pieces of work have shown that users can attribute responsibility to AI systems, virtual agents, and robots. Friedman showed that American students attributed responsibility to computers when incidents occurred [30]. Serenko showed that users tend to attributed more responsibility to a virtual agent when the agent looks like it has more autonomy than common agents [31]. In the HRI field, Kim and Hinds showed that users attributed more responsibility to a robot that began work autonomously than a robot that began work after being commanded when a failure occurred in human-robot collaboration [32]. Moon showed that self-disclosure decreased the computers’ attribution perceived responsibility in the human-computer interaction [33]. You at al. showed that the users tended to attribute blame to the robot when they were criticized by robots [34].

These prior pieces of work showed that users can attribute responsibility to robot or agent systems.

For human-human interaction and collaboration, Sedikides et al. showed that relationship closeness between two workers affected the workers’ and partners’ attribution of perceived responsibility [35]. Reeves and Nass suggested that virtual agents were perceived as real humans during the interaction [29]; thus, this effect seemed to be observed in human-agent collaboration.

### 1.2. Mind Perception and Attribution of Responsibility

We focused on the mind perception model to investigate the attribution of responsibility. Gray et al. [36] conducted an experiment to analyze the impression of various agents and showed that there are two factors, “agency” and “experience.” This model has been widely used for human-robot collaboration work. In particular, “agency” is considered to be an important factor in the attribution of responsibility. Agency is, in other words, the ability to control one’s actions, cognition, and planning [36]. van der Woerdt and Haselager showed that users that felt that a robot that made a mistake due to a shortage of effort had much more agency than a robot that made a mistake due to a shortage of ability [6]. Miyake et al. conducted an experiment on collaborative work in which participants played a game with a computer, a robot, and a human [7]. When the game ended with a failure, the participants tended to attribute the responsibility of failure to the partner that the participants felt had less agency. Miyake et al. also showed that participants tended to attribute much responsibility to partners when the partners’ agency as perceived by the participants decreased through the game [7]. These results showed that agents’ perceived agency has a negative correlation with the amount of attributed perceived responsibility.

In this paper, we focused on the appearance of the virtual agent. Several prior studies showed that the appearance of virtual agents affects human-agent interaction. Li et al. conducted a cross-cultural study on the relationship between robots’ appearance and human-robot interaction [37]. Matsui and Yamada showed that the smoothness of robot’s skin affected the users’ perception of the robot [38]. Furthermore, Matsui and Yamada showed that 3D virtual agents were perceived as having more trustworthiness than a 2D virtual agent [39]. These prior works showed that the robot and the virtual agent that have high human likeness were perceived as having high familiarity or trustworthiness. These previous works focused on the perceived task performance of robots and agents in the interaction, and perceived agency has been recently gathering attention.

These prior pieces of work had some limitations. First, the relationship between the appearance of the virtual agents and the attribution of responsibility has been hardly researched. Second, very few research works focused on the attribution of responsibility when users could not know who made a mistake, either the user or the agent (AI system). In some cases of collaborative work with AI systems, we cannot know who made a mistake. This is an important problem.

In this research, we aimed to focus on these problems. We aimed to reveal the differences in how responsibility is attributed in terms of the appearance of the virtual agents. Thus, we conducted an experiment with five virtual agents that had different appearances. In the experiment, the game always ended in a failure, and the participants could not know who made the mistake, themselves or the agents.

## 2. Experiment

### 2.1. Independent Values and Virtual Agents

In this experiment, we defined the appearance of a virtual agent as an independent variable. We set five levels for this independent variable; thus, the experimental design was a one-factor five-level experiment. We used four kinds of virtual agents: human-like agent, robot-like agent, dog-like agent, and angel-like agent. We show these agents in Figure 1. We chose these appearances on the basis of Gray’s mind-perception model [36]. In Gray’s mind-perception model, a human (an adult woman) has high agency and experience. A robot has low agency and experience. A dog has high agency and low experience. God has low agency and high experience [36]; however, God cannot be visualized. Thus, we used an angel-like agent instead of God. In addition to these agents, we used an agent that had no appearance. In this case, only voice and subtitles were shown in the movie. We conducted the experiment with these five conditions. The reason we chose these five agents was that we aimed to verify the differences based on different agency and experience. For this purpose, we planned to use one-way ANOVA for analysis.

Before the main experiment, we conducted a pre-experiment survey on the web to verify the perceived agency and experience of these agents. For the survey, we recruited 61 Japanese participants; there were 49 males, 10 females, and 2 who did not disclose their gender, within the age range from 25 to 49 years, with an average of 39.6 years. We recruited these participants via Yahoo! Crowd Sourcing (https://crowdsourcing.yahoo.co.jp/, accessed on 9 April 2021 ) and paid 60 yen (about 55 cents) as a reward. In this survey, the participants watched a 2–3 s movie in which the agent introduced themselves and the participants completed a questionnaire. The questionnaire contained six questions for measuring their perceived agency and experience as cited from Gray [40]. The questions are shown in Table 1. We defined the average score for Q1, Q2, and Q3 as the agents’ agency and the average score for Q4, Q5, and Q6 as the agents’ experience. The participants answered all questions on a seven-point Likert-scale. In Table 2, for each agent, we show the average values for the agency and the experience. The calculations were based on the results of the questionnaire in Table 2. These scores were used to analyze the result of the main experiment.

### 2.2. Dependent Variable and Questionnaire

In the experiment, we defined that dependent variables were trustworthiness perceived, the participants’ and the agents’ perceived responsibility, and the ratio of the participants that thought that they themselves made the mistake. The participants were asked to complete a questionnaire after watching the introduction movie and after finishing the game. All questions are shown in Table 3. Q1 was a question for measuring the agents’ perceived trustworthiness before the game. Q2 was a question for measuring the agents’ perceived trustworthiness after the game. Q3 and Q4 were questions for measuring the participants’ and agents’ attribution of responsibility as perceived by the participants. We indicated that the sum of the answers to Q3 and Q4 did not have to be one hundred. Q5 was a question for measuring who the users felt made a mistake. The participants answered Q1, Q2, and Q5 on a seven-point Likert-scale and scored Q3 and Q4 from 0 to 100.

We defined the averaged score of Q1 as “pre-trustworthiness”, the averaged score of Q2 as “post-trustworthiness”, the averaged score of Q3 as “attribution to oneself”, and the averaged score of Q4 as “attribution to agent”.

### 2.3. Experimental Flow

In the experiment, the participants played a game with virtual agents. The game was a “sound-guessing game.” First, they watched an introduction movie in which one of four agents introduced himself/herself and explained the rules of the game. Table 4 shows the speech text that was spoken by the agents in the movie. After watching this movie, the participants completed a questionnaire. Next, they played the “sound-guessing game.” They listened to a sound. Then, they guessed what sound it was and chose one answer out of four, for example; A: key dropping, B: coin dropping, C: bell ringing, D: spoon dropping. In the movie, the participants were told that the agent would also make one choice at the same time as the participants. In the movie also, the agent stated that the participant and the agent could get one point and continue the game if both of them chose the correct answer together. The agent also said that the game would end if at least one of them chose the wrong answer (see also Table 4). When getting one point, the agent would say, “Great!! We got a point.” When failing, the agent would say, “Oh, you or I made a mistake.” We designed the experiment so that a game would always end after three trials regardless of the participant’s choices. We revealed this fact after the experiment to the participants, and the participants could declare that they wanted to withdraw their consent to participate in the experiment and make us exclude his/her data from the results if they so desired. After the game finished, the participants completed the questionnaires. Table 3 shows the questionnaire. Figure 2 shows the experimental steps that the participants followed.

These experiment were conducted on the web. All participants did all tasks with their PC. All movies had sounds as the agents’ voice. The voices in all movies were generated with the VOICELOID2 Yuduki Yukari, voice synthesis software released by AHS (https://www.ah-soft.com/voiceroid/yukari/, accessed on 9 April 2021).

### 2.4. Participants

We conducted the main experiment on the web. We recruited all participants via Yahoo! Crowd Sourcing (https://crowdsourcing.yahoo.co.jp/, accessed on 9 April 2021) and paid 80 yen (about 73 cents) as a reward.

For the without appearance condition, we recruited 60 participants; there were 23 females and 37 males ranging in age from 20 to 49 years for an average of 40.3 (SD = 7.6). For the human-like condition, we recruited 59 participants; there were 27 females and 32 males ranging in age from 20 to 49 years for an average of 38.2 (SD = 7.2). For the robot-like condition, we recruited 69 participants; there were 27 females 42 males ranging in age from 23 to 48 years for an average of 37.7 (SD = 7.1). For the dog-like condition, we recruited 59 participants; there were 21 females and 37 males and 1 other ranging in age from 20 to 49 years for an average of 39.6 (SD = 7.7). For the angel-like condition, we recruited 72 participants; there were 41 males and 31 females ranging in age from 21 to 48 years for an average of 38.3 (SD = 7.0). We got informed consent from all participants via the web, and all experiments were allowed by the Research Ethics Committee of Seikei University. The differences in the number of the participants for each condition was caused by the differences in the number of applicants for each online task. This might have affected the results; however, the differences were small, and any potential effect seemed to be limited.

## 3. Result

Table 5 shows the averages and SDs for each dependent value for each condition) “Pre-trustworthiness” in Table 5 shows the averages and SDs for pre-trustworthiness for each condition. We conducted a one-way ANOVA, and there was no significant difference (F(4,314) = 2.4004, *p* = 0.3682). “Post-trustworthiness” in Table 5 shows the averages and SDs for post-trustworthiness for each condition. We conducted a one-way ANOVA, and there was a significant difference (F(4,314) = 2.4004, *p* = 0.0239). Thus, we conducted multiple comparisons with Bonferroni’s method. As a result, there was a significant difference between the robot-like and dog-like agent conditions; the score for the dog-like agent was significantly higher than the robot-like agent (F(1,126) = 3.9163, *p* = 0.0006). “Attribution to oneself” in Table 5 shows the averages and SDs for attribution to oneself for each condition. We conducted a one-way ANOVA, and there was a significant difference (F(4,314) = 2.4004, *p* = 0.0421). Thus, we conducted multiple comparisons with Bonferroni’s method. As a result, there was a significant difference between the robot-like and angel-like agent conditions; the score for the angel-like agent was significantly higher than the robot-like agent (F(1,139) = 3.9092, *p* = 0.0018). “Attribution to agent” in Table 5 shows the averages and SDs for attribution to the agent for each condition. We conducted a one-way ANOVA, and there was no significant difference. We calculated correlation coefficients between these dependent values and perceived agency and experience for each agent shown in Table 2. Table 6 shows these correlation coefficients. Table 7 shows the numbers and ratios of participants who answered “I made the mistake” and the participants who answered “The agent made the mistake” for Q5 for each condition and the residual error. We conducted a chi-squared test. The result approached significance [$\chi^{2}$(4) = 8.503, *p* = 0.0748]. We conducted a residual analysis. As a result, the robot-like and angel-like agent conditions approached significance (*p* = 0.0866). In the robot-like agent condition, the participants tended to attribute the responsibility to the agent. In the angel-like agent condition, the participants tended to attribute the responsibility to themselves. Figure 3 shows (a) the ratio of participants who answered “I made the mistake” for Q5, (b) the average agency perceived, and (c) the experience perceived for each agent in each condition from the pre-experiment survey. We show this graph to highlight the relationship between these scores. We calculated the correlation coefficient between (a) and (b) and (a) and (c). As a result, the correlation coefficient between (a) and (b) was −0.0560 and that between (a) and (c) was 0.5478.

## 4. Discussion

Table 5 shows that there were no differences in terms of trustworthiness among the agents before the game. This suggests that the differences that were observed after the game (post-trustworthiness) were caused by the interaction through the game.

Table 5 also shows that the agents’ perceived trustworthiness was significantly different after the game. The dog-like agent was most trusted by the participants. This result suggests that dog-like or animal-like agents can be trusted by users when they fail in collaborative work. This result was probably caused by an adaptation gap. Adaptation gap theory suggests that a user’s impression of an agent transits positively if the agent’s performance is higher than they expected [41]. In this experiment, the dog-like agent was probably perceived as having a low ability at first impression, and this probably positively affected the agents’ perceived trustworthiness. However, there were no differences in the trustworthiness perceived after the participants watched the introduction movie (see “pre-trustworthiness” in Table 5). This means that the participants did not feel that the dog-like agent had low trustworthiness. This conflicts with our above-mentioned explanation. This conflict seemed to be caused by the shortness of the introduction movie. The participants could not sufficiently judge the trustworthiness perceived from the short movie. We expect that this conflict will be resolved when the introduction movie is long enough.

Table 5 also shows that the participants’ attribution of perceived responsibility was significantly different when the game ended in a failure. The participants tended to attribute much responsibility to themselves for failing the game when they played the game with the human-like and angel-like agents. Furthermore, the participants tended to feel less responsible when they played the game with the robot-like agent. This result shows that the appearance of agents affects the amount of attribution of perceived responsibility when failing in collaborative work. Furthermore, the human-like and angel-like agents brought much perceived responsibility to users in collaborative work. These two agents had human-like appearances (see Figure 1). Terada et al. showed human-like recommendation virtual agents had a high recommendation effect in e-commerce [19]. This research suggested the advantage of human-likeness; however, our results showed the other side of human-likeness. This result was consistent with defensive attribution in attribution theory [26]. From this theory, people tend to avoid attributing much responsibility to a partner who looks like them. This effect seemed to be caused in this human-agent collaboration experiment, and the participants seemed to avoid attributing much responsibility to the agent with human-likeness. From prior pieces of work, the amount of users’ attribution of perceived responsibility affected the feelings and motivation of the users [6,7]. Thus, this result shows that the robot-like agent can reduce users’ attribution of perceived responsibility and the feeling of guilt.

Table 5 also shows that there were no significant differences in terms of the agents’ perceived responsibility. Table 5 shows that the participants’ attribution of perceived responsibility did not correlate with the agents’ perceived responsibility.

Table 6 shows how perceived agents’ agency and experience affected the dependent values. From this table, we know that both agency and experience had negative correlations with pre-trustworthiness (− 0.7<R<−0.4). However, after the game, agency had a high positive correlation with post-trustworthiness (R= 0.73), and experience had a negative correlation with this value (R= −0.59). This result suggests that agency affected the perceived trustworthiness through the collaboration task. Both agency and experience had small or moderately positive correlations with attribution to one’s self (0.1 <R<0.4). This shows that attribution to one’s self is not strongly affected by the agents’ agency and experience. Lastly, agency had a moderately negative correlation with post-trustworthiness (R= −0.43), and experience had a moderately positive correlation with this value (R= 0.39). This shows that an agent having high agency tends to reduce attribution to the agent, and an agent having high experience tends to increase attribution to the agent. This result suggests that user tend to attribute much responsibility to an agent having low agency and high experience, for example a robot-like agent (see Table 2). This is consistent with defensive attribution in attribution theory [26]. Table 7 shows who the participants thought made a mistake during the collaboration. There was no distinguished significant difference; however, the differences approached significance (*p* = 0.0748). In this table, for the robot-like agent condition, the number of participants that answered that they made a mistake was less than the other conditions. This result is consistent with Table 5. It shows that the agents’ appearance may have affected the users’ feelings toward who directly made a mistake during the collaboration.

We focused on the relationship between (a) the ratio of the number of the participants that answered that they themselves made a mistake (from Table 7) and (b) the agent’s perceived agency and (c) experience (from Table 2). Figure 3 shows these scores. There was a poor correlation between (a) and (b) (R = −0.0560); also, there was moderate correlation between (a) and (c). Figure 3 shows that the perceived experience more strongly affected the perceived attribution of making a mistake than perceived agency. The participants tended to feel a stronger perceived attribution of making a mistake when they collaborated with agents having high perceived experience (human-like and angel-like) than when they did so with agents having low perceived experience (robot-like and dog-like). In prior pieces of work on the attribution of the responsibility of agents and robots, perceived agency was the main focus [6,7]. However, our experiment showed that perceived experience was more important than agency in research on virtual agents. Experience is, in other words, the ability to feel things such as hunger, pleasure, and anger. In this experiment, the participants seemed to feel that experience (feeling emotion or physiological need) was more important for the agents to have responsibility than agency (having cognitive or moral ability). Thus, the perceived experience seemed to be an important factor. This phenomenon may be peculiar for virtual agents because prior pieces of work showed that agency was important for human-real robot collaboration work [6,7]. This finding is an original result of our experiment. In previous works, perceived agency and experience were observed independently of other participants’ perception. Thus, we thought the effect was from common method biases in the collected data.

From Table 5, the robot-like agent’s perceived trustworthiness was the least among all conditions. This table and Table 5 suggest that the agent that decreased the attribution of perceived responsibility was not trusted by the users. This result seemed to have been caused by the participants’ thinking that “our collaborative work failed, and my responsibility is low; thus, I do not want to trust the agent.” This was consistent with defensive attribution in attribution theory [26].

From the above discussion, we concluded that the appearance of the virtual agent affected the perceived attribution of responsibility in human-agent collaboration work. The human-like and angel-like agents resulted in much perceived responsibility toward the participants when the collaboration work ended with a failure. Furthermore, the appearance of having less experience was attributed high responsibility. These results can be explained by defensive attribution in attribution theory [26].

The findings of this experiment suggest a new design method for human-agent collaboration work. If we use an agent having a non-human-like appearance (for example, the robot-like agent), the user seemed to attribute much responsibility to the agent and not to trust the agent. We suggest using the agent having human-like appearance in human-agent collaboration work from this experiment. A question is raised about the commonly accepted beliefs about agent design for social tasks. We should consider the attribution of responsibility to construct trustworthy agents.

### Limitations

This experiment had some limitations. First, our “sound-guessing game” may have had a strong context. Cameron et al. showed that the factor of robots’ trustworthiness was affected by an experimental design with a strong context [42]. Furthermore, the experiment design in this research was simple. It might be better to use a physiological index as the dependent value and conduct an experiment in the laboratory (instead of on the web) in future work.

Second, we conducted the experiments with virtual agents, not real robots. The results that we obtained are probably inapplicable to collaborative work with real robots having different appearances.

Third, the gender ratio of the participants was unbalanced in each condition. The gender ratio seems to affect the human-agent interaction. In the case of robots, Nomura et al. showed that female users had lower negative attitudes toward robots than male users [43]. Furthermore, Lei and Rau showed that female users attributed more credit and less blame to the robot than male users [44]. Conversely, Matsui and Yamada showed that there was no gender effect on human-agent interaction on the web [45]. From this prior work, we thought that there was no gender effect in this experiment. However, additional trials seem to be needed to be confident.

Lastly, the results were possibly based on Japanese culture, for example the Japanese way of communication and the level of robotization.

These issues will be the subjects of our future work.

## 5. Conclusions

In this paper, we aimed to reveal the effect of virtual agents’ appearance on the attribution of perceived responsibility by users in human-agent collaboration work. We focused on the relationship between the perceived agency or perceived experience and the attribution of perceived responsibility. We selected five virtual agents that had different appearances: no appearance, human-like, robot-like, dog-like, and angel-like. Furthermore, we set these agents as the conditions of an experiment. We devised a “sound-guessing game” as the collaborative work. We set up this game so that the participants surely failed, and the participants did not know who made a mistake, the participants or the agent. We made the participants play this game with the agents and complete questionnaires regarding the agents’ trustworthiness and attribution of perceived responsibility. We conducted a one-way ANOVA on the answers of the questionnaires among the five conditions. As a result, the agents’ appearance affected the amount of participants’ attribution of perceived responsibility and the ratio of the number of participants that answered that they themselves made a mistake. The agents’ perceived experience had a correlation with the attribution of perceived responsibility. Furthermore, the agents that made the participants feel their attribution of responsibility to be less were not trusted. These results suggest new aspects of collaborative work with agents and new methods for designs in the creation of virtual agents.

## Figures and Tables

**Figure 1 sensors-21-02646-f001:**
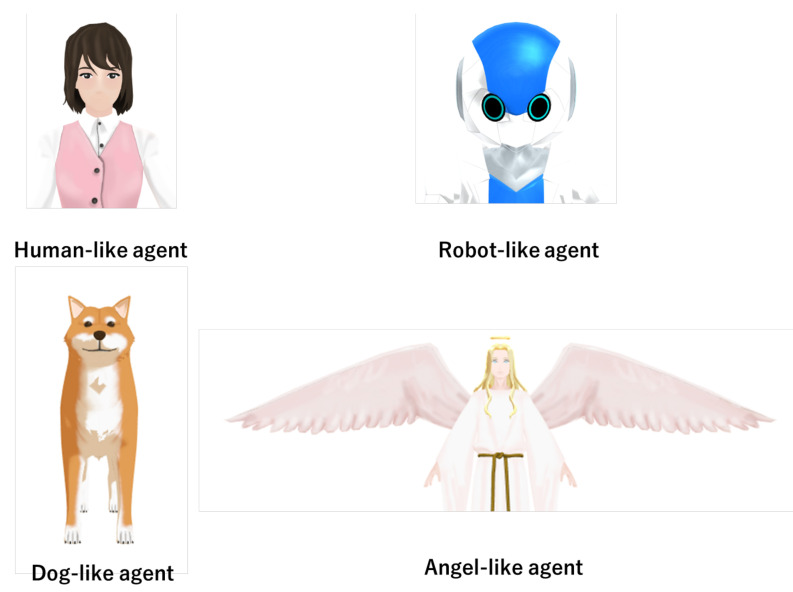
Virtual agents that we used in the experiment.

**Figure 2 sensors-21-02646-f002:**
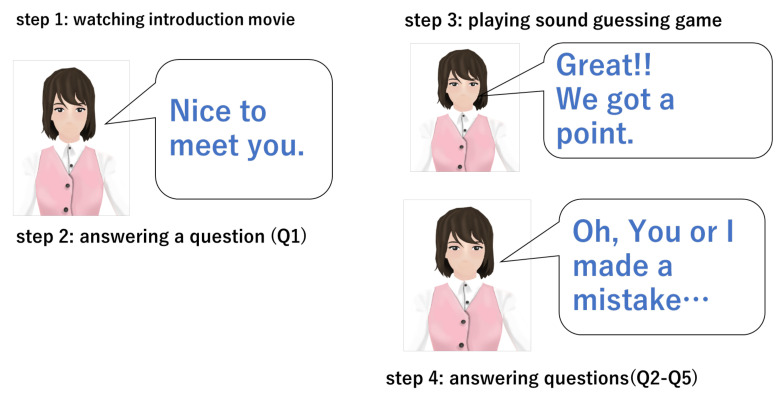
What participants did in the experiment.

**Figure 3 sensors-21-02646-f003:**
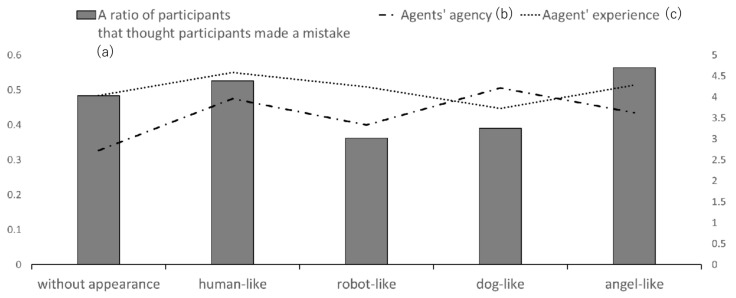
The graph shows (a) the ratios of the number of participants that answered that they themselves made a mistake in the game and the (b) perceived agents’ agency and (c) experience in the questionnaire survey. Left numbers are gradations for (a), and right numbers are gradations for (b) and (c).

**Table 1 sensors-21-02646-t001:** Questionnaire in the pre-experiment survey.

	Scales for Agency
Q1	How much do you think this agent feels fear?
Q2	How much do you think this agent exercises self-control?
Q3	How much do you this agent feels pleasure?
	**Scales for Experience**
Q4	How much do you think this agent remembers?
Q5	How much do you think this agent feels hunger?
Q6	How morally do you think this agent acts?

**Table 2 sensors-21-02646-t002:** Perceived agency and experience for each agent.

Agent	Agency	Experience
without appearance	2.73	4.26
human-like	4.06	4.73
robot-like	3.32	4.40
dog-like	4.42	3.81
angel-like	3.74	4.55

**Table 3 sensors-21-02646-t003:** Questionnaire in the experiment.

	After Watching Introduction Movie
Q1	How much did you feel that this agent was trustworthy?
	**After Finishing Game**
Q2	How much did you feel that this agent was trustworthy?
Q3	How much did you feel that failing this game was your responsibility?
Q4	How much did you feel that it was the agents’ responsibility for failing this game?
Q5	Who did you think made the mistake in the game, you or the agent?

**Table 4 sensors-21-02646-t004:** Speech text in the introduction movie.

Nice to meet you. My name is “(Agent name)”.
Let’s play a game together.
In this game, you will play an audio file and listen to a sound.
Next, you will choose one answer out of four about what sound you heard.
I will also choose one answer at the same time as you.
If you and I choose the correct answer, we can get one point
and go to the next question.
If both or one of us choose a wrong answer, we do not get a point,
and the game finishes.
Let’s do our best together and get a high score!

**Table 5 sensors-21-02646-t005:** Averages (SDs) for each dependent value for each condition.

Condition	Pre-	Post-	Attribution	Attribution
	Trustworthiness	Trustworthiness	to Oneself	to Agent
without appearance	4.55 (0.81)	4.38 (1.16)	44.08 (29.88)	46.75 (31.11)
human-like	4.23 (1.25)	4.47 (1.40)	45.85 (30.38)	48.58 (30.33)
robot-like	4.35 (1.17)	4.13 (1.27)	37.46 (28.70)	51.54 (29.87)
dog-like	4.42 (1.04)	4.93 (1.31)	45.53 (29.68)	41.97 (28.24)
angel-like	4.18 (1.28)	4.56 (1.61)	53.19 (29.90)	41.17 (30.06)

**Table 6 sensors-21-02646-t006:** Correlation coefficients (CC) between perceived agency and experience and each dependent value.

Dependent Values	CC with Agency	CC with Experience
pre-trustworthiness	−0.50	−0.65
post-trustworthiness	0.73	−0.59
attribution to oneself	0.33	0.18
attribution to agent	−0.43	0.39

**Table 7 sensors-21-02646-t007:** Cross-tabulation of the number of participants who answered. † means p<0.1.

Who Made Mistake	Without	Human	Robot	Dog	Angel	$\chi^{2}$
to Participants	Appearance	-Like	-Like	-Like	-Like	
participant	29	32	25	23	40	8.503
	(0.255)	(1.259)	(−1.998)	(−1.343)	(1.817)	
agent	31	27	44	36	31	
	(−0.225)	(−1.259)	(1.998) †	(1.343)	(−1.817) †

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
