# Peer review of "Who Is to Blame? The Appearance of Virtual Agents and the Attribution of Perceived Responsibility"

_sensors, 2021, doi:10.3390/s21082646_

Round 1

Reviewer 1 Report

In this work, authors aimed to reveal the effect of virtual agents’ appearance on the attribution of responsibility perceived by users in human-agent collaboration work. I have some comments which I would like to be addressed.

What was the key motivation behind focusing on the relationship between the perceived agency or perceived experience and the attribution of responsibility perceived?

Why authors choose five virtual agents?

Authors should further clarify and elaborate novelty in their abstract.

Explain the implications of your research.

Did authors checked the common method biased in the data?

What are the main research questions in the present work?

Below papers has some interesting implications that you could discuss in your introduction and how it relates to your work.

  • Anwar, Aizza, et al. "Impact of Music and Colour on Customers’ Emotional States: An Experimental Study of Online Store." Asian Journal of Business Research1 (2020): 104.
  • Wagner, Gerhard, Hanna Schramm-Klein, and Sascha Steinmann. "Online retailing across e-channels and e-channel touchpoints: Empirical studies of consumer behavior in the multichannel e-commerce environment." Journal of Business Research107 (2020): 256-270.
  • Ijaz, M.F.; Rhee, J. Constituents and Consequences of Online-Shopping in Sustainable E-Business: An Experimental Study of Online-Shopping Malls. Sustainability2018, 10, 3756.

Author Response

Thank you for inviting us to submit a revised draft of our manuscript.
We also appreciate the time and effort you and each of the reviewers have dedicated to providing insightful feedback on ways to strengthen our paper. Thus, it is with great pleasure that we resubmit our article for further consideration. We have incorporated changes that reflect the detailed suggestions you have graciously provided. We also hope that our edits and the responses we provide below satisfactorily address all the issues and concerns you and the reviewers have noted.

To facilitate your review of our revisions, the following is a point-by-point response to the questions and comments delivered in your review.

Your comment:
What was the key motivation behind focusing on the relationship between the perceived agency or perceived experience and the attribution of responsibility perceived?

Response:
Thank you for your comment.
Our key motivation is to investigate he relationship between agents' appearance and agents' attribution of responsibility on collaboration work.
Also for this purpose, we focused on two factors suggested by prior works that perceived by agents' appearance, "agency" and "experience".
We added these explanation in 5th and 6th paragraph in section 1.

Your comment:
Why authors choose five virtual agents?

Response:
Thank you for your comment.
We chose five agents based on the Gray's mind perception model.
Also we used five agents because we aimed to verify the differences based on different agency159and experience.
We added these explanation in first paragraph in section 2.1

Your comment:
Authors should further clarify and elaborate novelty in their abstract.

Response:
Thank you for your comment.
We added more ditails in abstract.

Your comment:
Explain the implications of your research.

Response:
Thank you for your comment.
We think that this research raise questions about commonly accepted beliefs of agents' design for social task.
We added this explanation in the last paragpaph in section 4.0.

Your comment:
Did authors checked the common method biased in the data?

Response:
Thank you for your comment.
We think that the effect of common method biases because perceived agency and experience were observed independently from other participants' perception in previous works.
We added this explanation in 6th paragraph in section 4.0.

Your comment:
What are the main research questions in the present work?

Response:
Thank you for your comment.
The present works focused on the perceived task performance of robots and agentsin interaction and perceived agency is recently gathering attention.
We added this expkanation in second paragraph in section 1.2.

Your comment:
Below papers has some interesting implications that you could discuss in your introduction and how it relates to your work.

Response:
Thank you for your suggestion.
We cited these papers in 4th paragraph in section 1.0

Reviewer 2 Report

Overall it is a very well written paper. I thoroughly enjoyed reading the work.

Couple of Comments/questions:

  • provide justification for why the sample size was different for different conditions, and did it have any impact on the results?
  • The justification for conflict in the trustworthiness of dog-like agent is not clear. Simply stating that this problem is for future is not acceptable.

Author Response

Thank you for inviting us to submit a revised draft of our manuscript.
We also appreciate the time and effort you and each of the reviewers have dedicated to providing insightful feedback on ways to strengthen our paper. Thus, it is with great pleasure that we resubmit our article for further consideration. We have incorporated changes that reflect the detailed suggestions you have graciously provided. We also hope that our edits and the responses we provide below satisfactorily address all the issues and concerns you and the reviewers have noted.

To facilitate your review of our revisions, the following is a point-by-point response to the questions and comments delivered in your review.

Your comment:
provide justification for why the sample size was different for different conditions, and did it have any impact on the results?

Response:
Thank you for your suggestion.
The differences of the number of the participants for each condition was caused by the differences of the number of applicants for each online task. This might affect the results, however, the differences were small and potential effect seemed to be limited.
We added this explanation in the last paragraph in section 2.4.

Your comment:
The justification for conflict in the trustworthiness of dog-like agent is not clear. Simply stating that this problem is for future is not acceptable.

Response:
Thank you for your comment.
We think that this conflict seemed to be caused by the shortness of the introduction movie. The participants could not sufficiently judge the trustworthiness perceived from shot movie. We expect that this conflict will be resolved when the introduction movie is enough long.
We added this explanation in the second paragraph in section 4.0.

Reviewer 3 Report

This is an interesting study which investigates the trustworthiness of the agents based on their appearance. Thee method seems to be well-designed, the size of the sample is satisfactory, the methods are appropriate and the conclusions are sound.

I would add as a limitation the fact that the participants in the experiment were only of Japanese nationality. In my opinion, this is a limitation because (1) Japanese way of communication is much more based on visual elements than the one in other nations, and (2) the level of robotization in Japan is quite high with respect to other countries. This does not diminish the contributions of the work, however.

The main problem of the paper is the abundance of language errors - grammatical, stylistic, and punctuational, so that it is difficult to read it. It needs a significant editing. Below are some of the linguistic issues (the list is not exhaustive):

lines 73-74) We focused on responsibility for when only collaborative work fails in human-agent collaboration -> unclear sentence and repetition (collaborative, collaboration)

lines 84-85) than a robot that begins work after being commanded after a failure in human-robot collaboration -> unclear sentence and repetition (after, after)

lines 96-97) We focused on the mind perception model to think about the attribution of responsibility -> We focused on the mind perception model to investigate the attribution of responsibility

lines 105-106) the game ended in failure, -> the game ended with a failure,

lines 112-113) Many prior researches showed -> Several prior studies showed

line 117) the 3D virtual agent were -> the 3D virtual agents were

lines 117-118) than 2D virtual agent[35] -> than the 2D virtual agent [35]

multiple places in the article: space should be left between the opening bracket of the cited paper and the word before it, as in the previous example 

line 118) that the robot and virtual agent that have -> that the robot and the virtual agent that have

line 133) appearance of virtual agent -> appearance of the virtual agent

lines 146-147) and 2 who identified as “other" ranging in age from 25 to 49 years for an average of 39.6 -> and 2 who did not disclose their gender, within the age range from 25 to 49 years, with an average of 39.6 years

lines 155-156) We show the averages for agency and experience for each agent from this questionnaire survey in Table 2. -> In Table 2, for each agent we show the average values for the agency and the  experience. The calculations are based on the results of the questionnaire in Table 2.

line 194) This experiment were conducted -> These experiments were conducted

line 211) Committee of XXXX University -> Should be corrected

line 358) These are our future work. -> These issues will be a subject of our future work.

Author Response

Thank you for inviting us to submit a revised draft of our manuscript.
We also appreciate the time and effort you and each of the reviewers have dedicated to providing insightful feedback on ways to strengthen our paper. Thus, it is with great pleasure that we resubmit our article for further consideration. We have incorporated changes that reflect the detailed suggestions you have graciously provided. We also hope that our edits and the responses we provide below satisfactorily address all the issues and concerns you and the reviewers have noted.

To facilitate your review of our revisions, the following is a point-by-point response to the questions and comments delivered in your review.

Your comment:
I would add as a limitation the fact that the participants in the experiment were only of Japanese nationality. 

Response:
Thank you for your comment.
We agree with your opinion.
Thus added this problem in limitation section, section 4.1

Your comment:
The main problem of the paper is the abundance of language errors - grammatical, stylistic, and punctuational, so that it is difficult to read it. It needs a significant editing. Below are some of the linguistic issues (the list is not exhaustive):

Response:
Thank you for your comment.
We modified the issues that you pointed out.
We get English proofreading for this manuscript by native.

Round 2

Reviewer 1 Report

.